# Future Climate Effects on Basal Stem Rot of Conventional and Modified Oil Palm in Indonesia and Thailand

Robert Russell Monteith Paterson [1,2]

1 Department of Biological Engineering, Gualtar Campus, University of Minho, 4710-057 Braga, Portugal; russell.paterson@deb.uminho.pt
2 Department of Plant Protection, Faculty of Agriculture, Universiti Putra Malaysia, Serdang 43400, Selangor, Malaysia

**Abstract:** Oil palms (OP) produce palm oil, a unique commodity without commercial alternatives. A serious disease of OP is basal stem rot (BSR) caused by *Ganoderma boninense* Pat. Climate change will likely increase BSR, thereby causing mortality of OP and reduced yields of palm oil. Work is being undertaken to produce modified OP (mOP) to resist BSR, although this will take decades for full development, if successfully produced at all. mOP will not be 100% effective, and it would be useful to know the effect of mOP on the key parameters of BSR incidence, OP mortality, and yield loss. The current paper employed CLIMEX modeling of suitable climates for OP and modeling narratives for Indonesia and Thailand. Indonesia is the largest producer of OP and Thailand is a much smaller manufacturer, and it was informative to compare these two countries. The gains from using mOP were substantial compared to the current production of some other continents and countries. The current paper, for the first time, assessed how climate change will affect BSR parameters for conventional and mOP. Greater consideration of the potential benefits of mOP is required to justify investing in the technology.

**Keywords:** *Elaeis guineensis*; *Ganoderma boninense*; modeling; climate emergency; palm oil; SE Asia

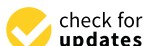



## 1. Introduction

Palms are paradigmatic plants. Oil palms (OP) are perhaps the most economically significant palm and form vast plantation forests in many tropical regions [1]. There is only limited information about the effect of climate change on tropical plants, including crops [2]. Palm oil is a very important commodity [1] and is used in cooking, food, cosmetics, plastics, chemicals, biodiesel, and detergents [3]. The economy of Indonesia is supported by the income generated from the industry, and the country is the largest producer of palm oil. Thailand is a significant producer and is located close to Indonesia, which is instructive for comparative purposes. However, OP are associated with detrimental conversion of peat soils and deforestation, which contributes to climate change and smog, with negative effects on biodiversity [4]. Deforestation has been implicated in zoonotic diseases, through increasing the contact of humans with disease-carrying animals [5].

Climate change will severely affect OP production after 2050 [6–8], and land cover and management issues are relevant to plantations [3]. Palm oil demand has transformed these topics by greatly increasing deforestation by the creation of more plantations. Climate change in the 21st century will reduce the sustainability of plantations, and has serious implications for biodiversity loss, food security, and the economics in palm oil producing, and other, countries [9–13]. Palm oil currently accounts for more than 30% of global demand for vegetable oil, and 61% of palm oil comes from Indonesia, where 65, 25, and 3% of OP plantations were located in Sumatra, Kalimantan, and Sulawesi, respectively, in 2009. The remainder is in other regions, including Papua and Java [14]. Lam et al. [6] indicated that the greatest potential for developing new plantations is in Papua and Sulawesi. In addition, Thailand produces a significant volume of palm oil [1]. The incentive for developing new

plantations must be tempered by the environmental requirement to preserve natural forest and reduce climate change [4].

Furthermore, basal stem rot (BSR) of OP, a serious disease caused by the fungus *Ganoderma boninense* Pat., is likely to increase, making the industry much less sustainable in SE Asia [9–12], and which has severely affected the industry in Indonesia and Thailand over the past 100 years. The disease causes losses of USD 50 to 350 million per annum [15,16], and this represents 0.1 to 0.7% of the total value of the industry [3]. Currently, BSR can kill 80% of a plantation at 50% of their economic lifespan [1] and can reduce yields by 50%–80%. High incidence in individual plantations is devastating, and BSR has caused severe economic losses in Malaysia [17], which will be compounded as BSR increases [9,10,18]. Incidentally, there have been approximately 15 similar reviews of BSR published between 2021 and 2023 (SCOPUS) (e.g., references [19–22]), to which the reader can refer, with the later papers often repeating information provided in the earlier ones. In terms of a review of new information, Bharuden et al. [22] discussed the current genomics/transcriptomics research relating to the disease, which is useful.

In general, the current climate in Indonesia and many other SE Asian countries, is highly suitable for growing OP [23], implying that the plant will have maximum resistance to BSR compared to the situation in less suitable climates, and yet the incidence is high and increasing in many regions. The disease was first described in 1931 in Malaysian plantations and found subsequently over all SE Asia [24], where the incidence was 50% of OP in many areas and was considered very serious, even in 1965 [25]. BSR can kill 80% of the palms midway through their economic life span [1] and ground level OP are especially prone to the disease, while *G. boninense* can be present in the plant, despite symptoms being absent [1,9]. Furthermore, young, mature, and seedling OP can be affected by BSR, thereby reducing yields by ca. 65% [1,9].

Future climate effects on OP growth, yields, and BSR incidence indicate that Thailand will be more seriously affected than Indonesia [7,8,11,12,26]. Currently, the disease increases over a 25-year generation period [1], in which time the climate could have changed to favor BSR. The species of *Ganoderma* that cause BSR are difficult to determine, as species concepts are poorly described [27]; although *G. boninense* is currently applied to the fungus that infects OP in SE Asia. A distinct taxonomic group was detected within "cluster D.4.2" (Fryssouli et al. [28]), which corresponded to *G. boninense* in an analysis of ITS rDNA of specimens, including some from Indonesia and Thailand. Also, there was a related taxon of Indonesian specimens designated "D3", making the situation more complex. The taxonomic situation becomes more complicated still when the enormous number of specimens within plantations is considered, compared to the few that have been studied.

The variation within *Ganoderma* specimens from OP may be sufficient to permit a high level of adaption to climate change. The fungus reproduces sexually, allowing even more genetic variation to occur, and it produces millions of spores from even one basidiome, thereby permitting further variation [22]. Increased complexity in population structure and evolution in *G. boninense* will occur the most where the longest interactions with OP have occurred, as in Malaysia and Indonesia, and when it is adapting to environmental changes [29]. The heterogeneous nature of the fungus will enable it to adapt to climate change more readily than OP through the selection of more virulent taxa [16,29,30] and will likely become more virulent with the changing climate.

Furthermore, climate change will likely increase BSR incidence, due to stress on OP [16,31]. OP have adapted to SE Asia from Africa; but this was assisted by farming, selection, and propagation methods [1], which have obviously not occurred with the fungus. The incidence of *G. boninense* will increase with climate change, due to greater change occurring in the fungus compared to OP [10]. On the other hand, farming, selection, and propagation methods could assist OP to resist climate change, together with the various amelioration methods that could be applied [3]. The fungus has already adapted to infecting jungle/forest trees, coconut, rubber, and OP [17], indicating it has an inherent degree of adaptability, which may extend to adapting to climate change. The OP industry

was developed later in Thailand, compared to Indonesia and Malaysia, and experiences currently less infection [12].

The El Nino event in Malaysia in 1997 contributed to thick haze, high temperature stressing of OP, and increased incidence of BSR [31], and a similar situation would have occurred in Indonesia and other OP growing countries, such as Thailand. In Colombia, the OP yield was reduced by the El Niño and La Niña climate events [10], not least because OP are highly susceptible to drought [4]. Drought causes more disease in OP [4,23] including greater incidence of Fusarium wilt [1] for example.

There are four scenarios to consider for BSR of OP under the influence of climate change. First, the climate remains highly suitable for OP in certain regions, implying high resistance to *G. boninense*; second, more BSR occurs when the climate becomes less suitable for OP; third, the climate becomes unsuitable for OP, with *G. boninense* causing increased disease; and fourth, the climate becomes suitable for OP growth in new regions, allowing lower susceptibility to BSR [10].

BSR detection methods have remained unchanged for decades, and the introduction of Agriculture 4.0 technology may improve the situation [32], whereby simulation modeling and "big data" sets [7,8] will become pivotal to the new era [33]. Modern climate computer models have proven their suitability for OP growth information, indicate that growth will be affected detrimentally until the year 2100 [7,8]. These models have already been employed to provide disease assessments for OP [9–12]. OP management, spatial planning, and sustainable development relate climate to OP growth and disease, and the interfaces between dynamic models and social-ecological planning of OP landscapes are also germane.

Biotechnology can improve plant breeding by introducing novel genotypes that may result in improvement in palm oil production [34]. High level genetic breeding to create modified OP (mOP) requires multidisciplinary and collaborative research at a high level. Selecting for 100% resistance results in high selection pressures for new variants of pests/pathogens, which can overcome the desired resistance [35]. Nevertheless, the initial stages are underway of creating mOP resistant to inclement climates and the ability to combat BSR [36–38]. Resistant OP may overcome unfavorable growth conditions [39]. Zhou et al. [26] detected the myeloblastosis (MYB) gene family in OP, the largest and most abundant transcription factor families in plants, which have vital roles in development, abiotic stress response, hormone signal transduction, and secondary metabolite regulation. However, it will be decades before these results will be translated into producing a mOP available for planting on a commercial basis. Furthermore, Paterson et al. [40] discussed producing mOP with a high lignin content to control BSR, and the concept was investigated further in reference [41]. Phenylalanine ammonia lyase and cinnamyl alcohol dehydrogenase, in particular, may be targets for upregulation in OP, ideally to produce syringyl-enriched defense lignin in OP. Lanosterol 14$\alpha$-Demethylase [38], a key enzyme involved in the biosynthesis of ergosterol, was an early target for developing BSR resistant OP. Developing mOP resistant to BSR has begun, but whether they are the solution to controlling the disease under climate change remains speculative.

An assessment has been published concerning how future climates for growing conventional OP (cOP) and mOP will affect mortality and palm oil yields in Indonesia, Thailand, Malaysia, and Papua New Guinea (PNG) [42]. The improvements from using mOP were not dramatic but appeared most necessary for Thailand compared to the other countries. The improvements in mortality and yields in Indonesia and Malaysia were large when compared to the number of OP and yields of palm oil in the other countries.

The present study considers, for the first time, how future climate will affect (a) BSR, (b) mortality of OP from BSR disease, and (c) changes in palm oil yield from BSR in cOP and mOP. This will assist in assessing the benefits of mOP in Indonesia and Thailand. Indonesia was considered because it is the largest palm oil producer. In addition, the current and future climates are similar in Malaysia and Indonesia, whereas those for Thailand are different [42], hence providing an interesting contrast. The information will be of

considerable utility to researchers, OP plantation managers, environmentalists, economists, and politicians, and the conclusions are applicable to other OP growing countries.

## 2. Materials and Methods

The Paterson et al. [7,8] models provided schemes for future suitable climates for growing OP. Paterson et al. (a) [7] and (b) [8] provided maps of (a) Indonesia for current time, 2050, and 2100, and (b) Indonesia and Thailand for current time, 2030, 2070, and 2100. The magnification of maps to focus on these countries used the standard magnification facility of a personal computer. Climates were assessed visually, to provide percentage suitabilities from the designated colors of the maps, where each color represented a particular climate suitability.

The distribution model for OP under climate scenarios was developed using CLIMEX for Windows, Version 347 (Hearne Scientific Software Pty Ltd., Melbourne, Australia, 2007). Climate data and climate change scenarios were determined using CliMond $10'$ gridded climate data. The potential future climate was characterized using A1B and A2 SRES scenarios [7], available from the CliMond dataset and fitting of CLIMEX parameters employing the Global Biodiversity Information Facility. Global distribution data of OP were used in parameter fitting, where 124 records were used and SE Asian distribution data were reserved for the validation of the model. The OP distribution was determined using the Global Biodiversity Information Facility (GBIF) (http://www.gbif.org/, accessed on 9 November 2015) and additional literature on the species in CAB Direct (http://www.cabdirect.org/web/about.html, accessed on 9 October 2015). The collection of data was based also on the distribution of *Elaeis guineensis* Jacq. in reference [8], with 2465 records utilized in fitting the parameters. CLIMEX was used with the A2 Special Report on Emissions Scenarios (SRES) scenario. A mechanistic niche model using CLIMEX software (Version 347) supported ecological research, incorporating the modeling of species' potential distributions under differing climate scenarios and assuming that climate was the paramount determining factor of plant distributions. The CLIMEX output categorized areas according to climate, designating highly suitable, suitable, marginal, and unsuitable climates, based on other studies conducted through CLIMEX.

### 2.1. Basal Stem Rot Incidence

The BSR incidences of cOP in this report were based on published climate suitability maps [7,8] and were taken from references [11,12], and the details are not repeated herein. To determine the incidence of control of BSR from planting mOP, the introduction of mOP to plantations was designated to begin from after 2050 and that 10% mOP would be planted every 5 years thereafter; hence, 40% mOP would be planted from 2050 to 2070 and 60% from 2070 to 2100. The efficiency of control was taken as 90%. The values for incidences of BSR from planting mOP were subtracted from the incidences for cOP in 2070 and 2100.

### 2.2. Mortality from Basal Stem Rot

Mortalities of OP were designated as 50% of those infected by BSR, a figure taken from the literature (see Introduction and reference [1]).

### 2.3. Yield of Palm Oil Resulting from Basal Stem Rot

Obviously, the yield for those OP that died was zero. Therefore, if the mortality was 20%, then the yield reduction was 20%. For the remaining living OP, BSR incidence was determined as above, and a 50% yield reduction was assumed for these (see Introduction and reference [1]). Hence, a 50% BSR incidence was equivalent to a 25% yield reduction. The yield reduction for the dead OP was added to the yield reduction for the infected OP, which was subtracted from a nominal 100% at the current time.

### 2.4. Determination of the Numbers of Oil Palm Affected and Yield Values

These were taken from data in reference [1].

*2.5. Costs*

The costs were based on those for BSR in Malaysia [15], where these estimates have been made, and they are not available for other countries. The incidence for Malaysia was used to give a unit cost for infection based on Malaysian data, which was then multiplied by the incidence in either Indonesia or Thailand. The number of OP in Malaysia was used to give a standard value, which was multiplied by the number in Indonesia or Thailand.

Costs in Indonesia and Thailand were calculated using the following equations:

1.    Cost from infection rate per year

$$A = (B \div C) \times D$$

A is cost per infection rate; B is current cost of infection in Malaysia ($365 \times 10^6$) [15]; C is BSR incidence in Malaysia (29%) [12]. D is incidence in Indonesia or Thailand.

2.    Cost from number of oil palms

$$X = (A \div E) \times F$$

X is cost to Indonesia or Thailand; A is cost factor per infection rate; E is the number of OP in Malaysia ($61.0 \times 10^7$); F is the number of OP in Indonesia or Thailand for a particular time.

**3. Results**

Figure 1 indicates the percentage incidences of BSR in Thailand and Indonesia. The initial incidences in Indonesia and Thailand were 20 and 10%, respectively, and the increase in BSR was considerably less in Indonesia than in Thailand. The percentages increased at the same rate for cOP and mOP until after 2050 for both Indonesia and Thailand. The increases for Indonesia in 2070 were to 42% and 36% for cOP and mOP, respectively and the equivalent figures for 2100 were 70% and 53%. The data for Thailand were (a) 62% and 52% for cOP and mOP by 2070 and (b) 98% and 71% for cOP and mOP by 2100.

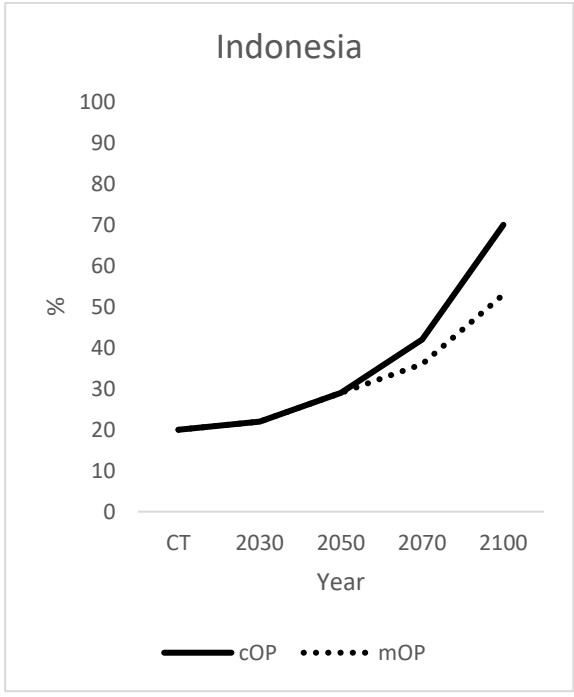 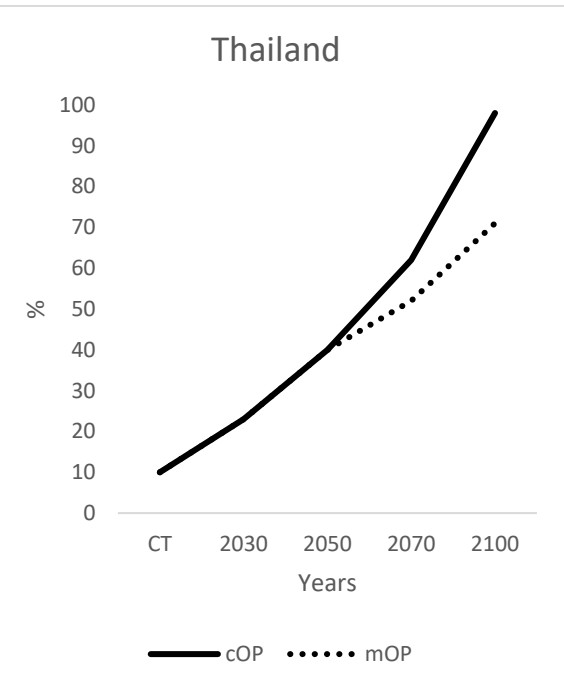

**Figure 1.** Percentage basal stem rot of conventional OP (cOP) and modified OP (mOP) influenced by future climate. CT = current time.

The total numbers of OP with BSR are presented in Figure 2. The total numbers for the current time were $1.8 \times 10^8$ and $9.0 \times 10^6$ for Indonesia and Thailand, respectively. The increase in the numbers of OP with BSR were considerably less for Indonesia than Thailand after 2030. The numbers of infected OP for Indonesia in (a) 2070 were $3.8 \times 10^8$ and $3.3 \times 10^8$ for cOP and mOP, respectively, and (b) 2100 were $6.4 \times 10^8$ and $4.8 \times 10^8$ for cOP and mOP, respectively. The number of infected palms in Thailand in (a) 2070 were $5.6 \times 10^7$ and $4.7 \times 10^7$ for cOP and mOP, respectively, and (b) 2100 were $8.8 \times 10^7$ and $6.4 \times 10^7$ for cOP and mOP, respectively.

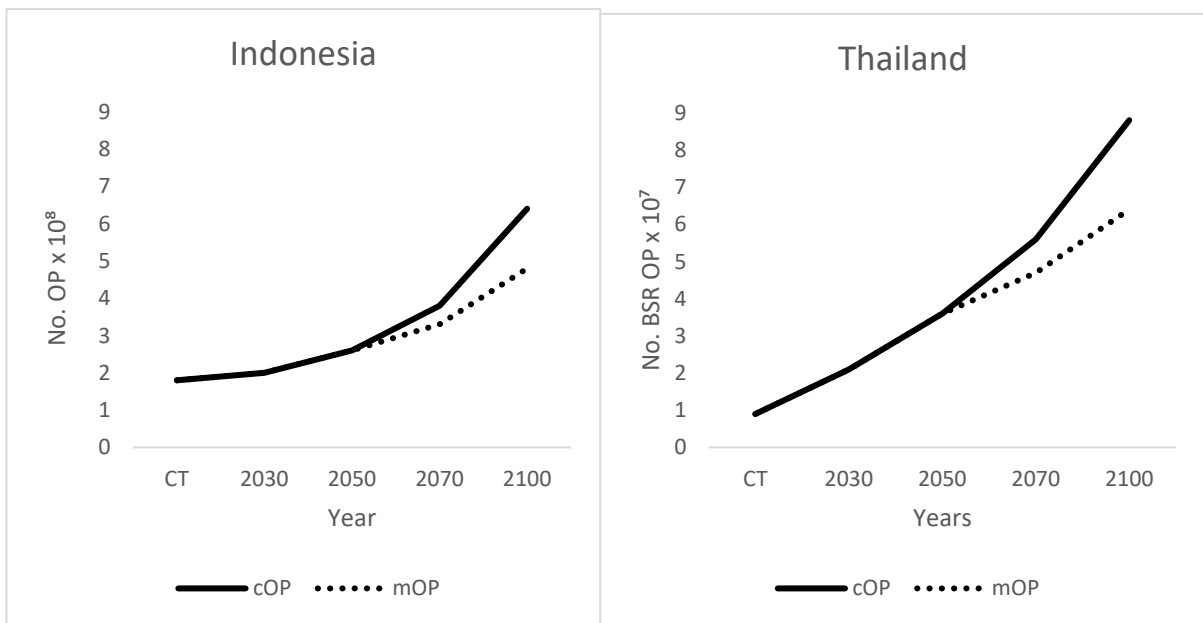

**Figure 2.** Total numbers of conventional OP (cOP) and modified oil palms (mOP) with basal stem rot, as influenced by future climate. CT = current time. Total number of OP = $9.1 \times 10^8$ for Indonesia and $9.0 \times 10^7$ for Thailand [1].

The equivalent values for the percentage mortalities of OP from BSR infection are provided in Figure 3.

The initial mortalities for OP were 10 and 5% for Indonesia and Thailand, respectively. The increase in mortalities for the subsequent years was considerably less for Indonesia. In Indonesia, the values for 2070 were (a) 20 and 18% for cOP and mOP, and (b) 35 and 26% in 2100 for cOP and mOP. For Thailand, the values for 2070 were 31 and 26% for cOP and mOP, respectively. For 2100, the Thailand data were 49 and 36% for cOP and mOP, respectively.

In Figure 4, the numbers of dead OP from BSR infection are demonstrated. The initial number of deaths from BSR were $9.0 \times 10^7$ and $5.0 \times 10^6$ for Indonesia and Thailand, respectively. Subsequent deaths increased more slowly for Indonesian OP. By 2070, the mortalities were $1.9 \times 10^8$ and $1.6 \times 10^8$ for cOP and mOP, respectively for Indonesia. These increased to $3.2 \times 10^8$ and $2.4 \times 10^8$ for cOP and mOP, in 2100. The figures for 2070 in Thailand were $2.8 \times 10^7$ and $2.4 \times 10^7$ for cOP and mOP, and for 2100 the equivalent figures were $4.4 \times 10^7$ and $3.2 \times 10^7$.

The percentage yields for Indonesia and Thailand are presented in Figure 5. There was a slow decrease in yield for Indonesia to 91% for cOP until 2050. This decreased rapidly to 85% by 2070 and to 49% by 2100. The equivalent figures for Thailand were 73, 56, and 50% for 2050, 2070, and 2100, respectively. The mOP yields decreased less quickly for Indonesia to 85% in 2070 and 72% in 2100. The mOP were less effective for Thailand in ameliorating the decrease in yields (Figure 5). The yields of palm oil were much higher in Indonesia than Thailand, with initial values of $3 \times 10^7$ t yr$^{-1}$ and $1.7 \times 10^6$ t yr$^{-1}$, respectively (Figure 6). The yields decreased more slowly for Indonesia until 2070, reaching $2.5 \times 10^7$ t yr$^{-1}$ for

Indonesia and $9 \times 10^5$ t yr$^{-1}$ for Thailand. There was a dramatic decrease for Indonesian cOP in 2100 to $1.5 \times 10^7$ t yr$^{-1}$, with the 2100 value for Thailand being $7 \times 10^5$ t yr$^{-1}$. The use of mOP in Indonesia in 2100 made a large difference at $2.1 \times 10^7$ t yr$^{-1}$, although a less significant increase was observed for Thailand at $8.0 \times 10^5$ t yr$^{-1}$.

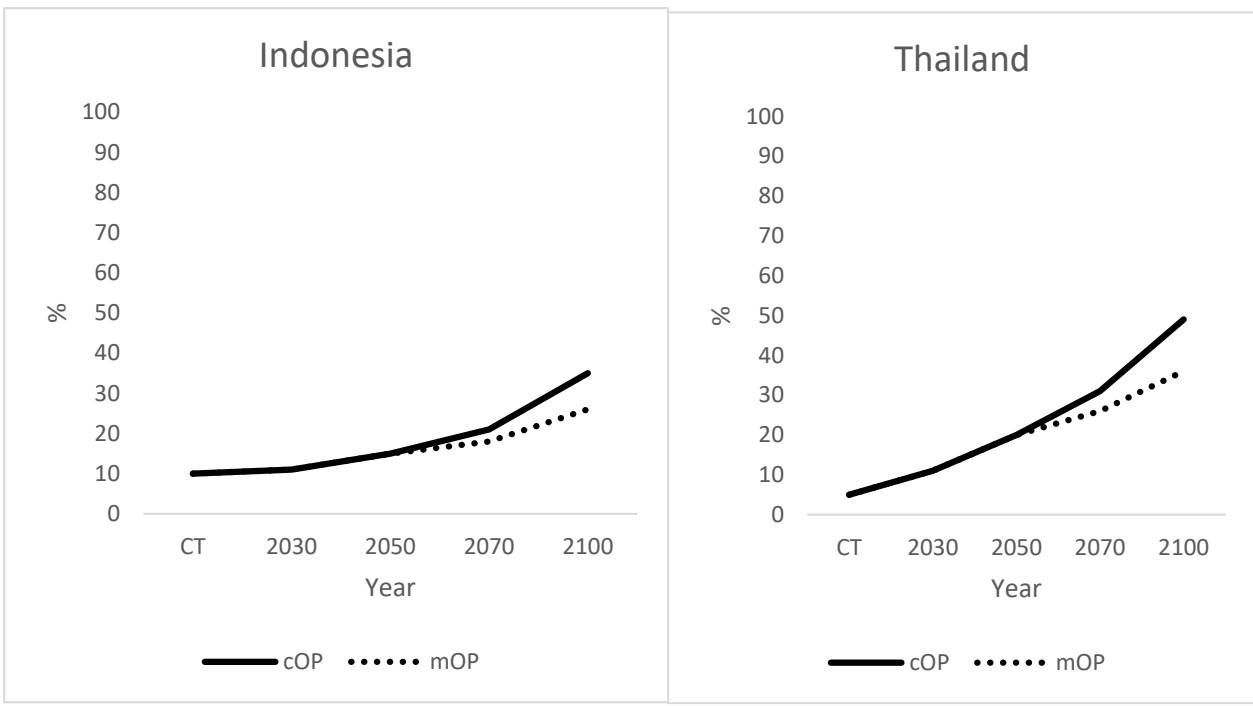

**Figure 3.** Percentage mortalities of conventional (cOP) and modified oil palm (mOP) from basal stem rot in response to future climate. CT = current time.

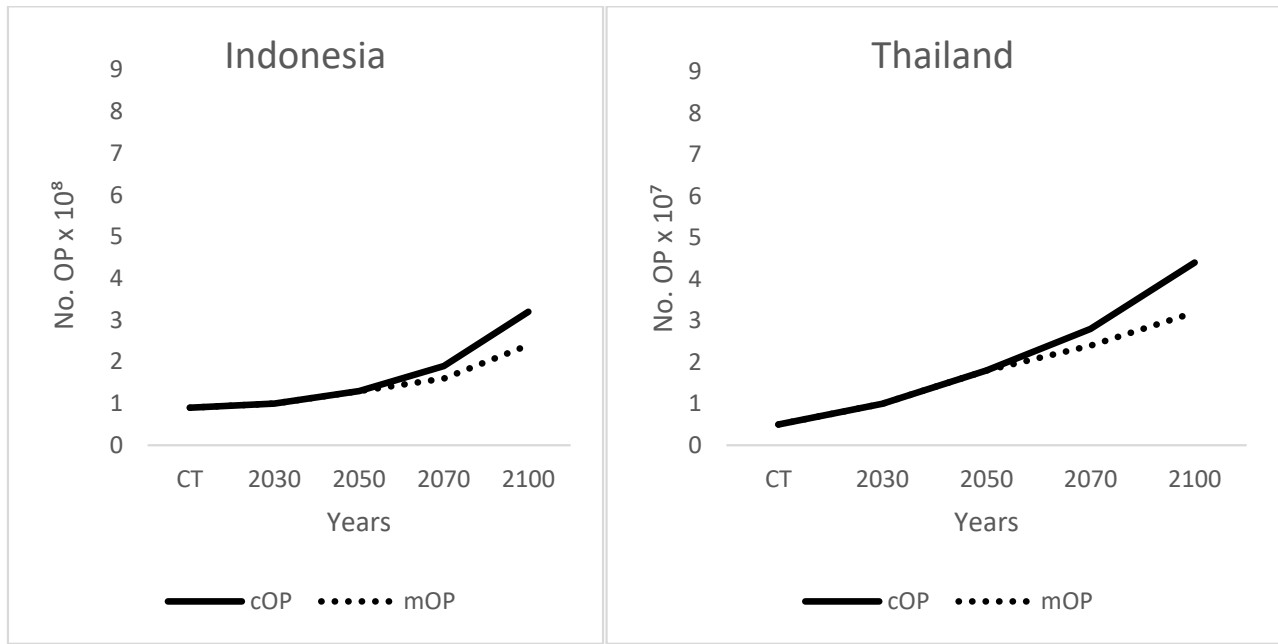

**Figure 4.** Number of conventional (cOP) and modified oil palm (mOP) killed by basal stem rot. CT = current time.

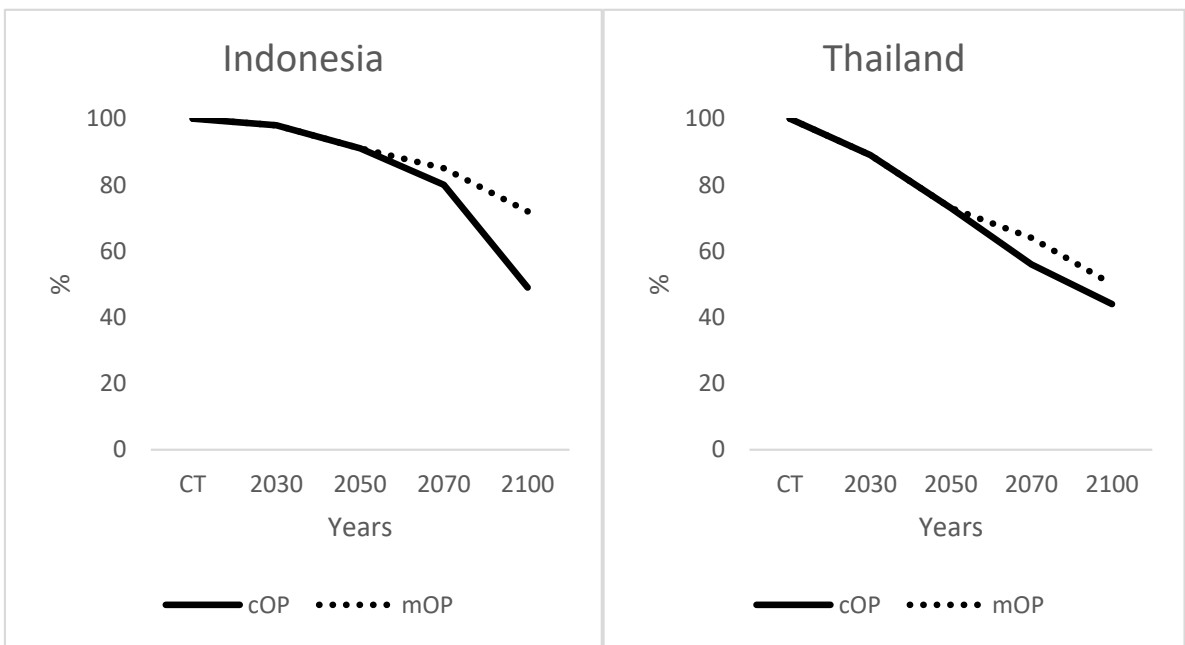

**Figure 5.** Percentage yield decrease from basal stem rot in response to future climate. CT = current time.

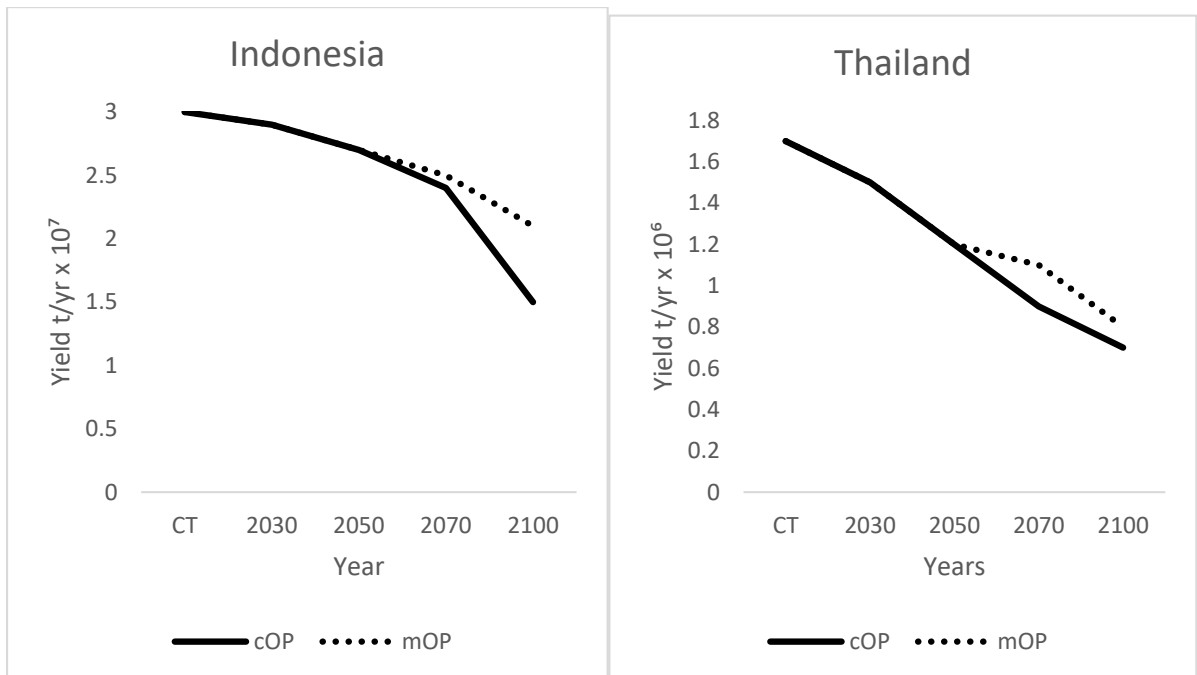

**Figure 6.** Yields (t yr$^{-1}$) of conventional palm oil (cOP) and modified oil palm (mOP) in response to basal stem rot for future climate. Yields were obtained from reference [1]. CT = current time.

The numbers of OP saved from BSR by planting mOP in Indonesia for 2070 and 2100 were $5.0 \times 10^7$ and $1.6 \times 10^8$, respectively (Table 1). For Thailand, the numbers saved from BSR in 2070 and 2100 were $1.0 \times 10^7$ and $3.0 \times 10^7$, respectively. The numbers of OP saved from death because of BSR in Indonesia in 2070 and 2100 were $3.0 \times 10^7$ and $8.0 \times 10^7$, respectively. For Thailand, the equivalent numbers were both $1.0 \times 10^7$ (Table 1). The yield improvements from using mOP in Indonesia for 2070 and 2100 were $1.0 \times 10^6$ t yr$^{-1}$ and $6.0 \times 10^6$ t yr$^{-1}$, respectively, and for Thailand, the equivalent yields saved were both $1.0 \times 10^5$ t yr$^{-1}$.

**Table 1.** A. BSR of OP: Number ($\times 10^7$) of oil palm (OP) saved from basal stem rot (BSR) as affected by future climate by planting modified oil palm (mOP). B. Mortality from BSR: Number ($\times 10^7$) of OP saved from death from BSR as affected by future climate by employing mOP. C. Yields after BSR infection: savings in yield (t yr$^{-1}$ $\times 10^5$) from employing mOP in relation to BSR as affected by future climate.

| | Year | |
|---|---|---|
| | **2070** | **2100** |
| **A. BSR of OP** | **Number of OP** | **Number of OP** |
| *Indonesia* | 5.0 | 16.0 |
| *Thailand* | 1.0 | 3.0 |
| **B. Mortality from BSR** | **Number of OP** | **Number of OP** |
| *Indonesia* | 3.0 | 8.0 |
| *Thailand* | 1.0 | 1.0 |
| **C. Yields after BSR infection** | **Yield** | **Yield** |
| *Indonesia* | 10.0 | 60.0 |
| *Thailand* | 1.0 | 1.0 |

The costs from employing mOP compared to the costs for using cOP in Indonesia and Thailand are provided in Table 2, together with the potential savings from planting mOP. The current time costs to Indonesia were US$ $376 \times 10^6$ and for Thailand US$ $19 \times 10^6$. The savings from planting mOP were US$ $113 \times 10^6$ for Indonesia in 2070 and US$ $319 \times 10^6$ by 2100. The equivalent figures for Thailand were $18 \times 10^6$ and $50 \times 10^6$ US$.

**Table 2.** Costs and savings (US$ $\times 10^6$) entailed from planting modified oil palm (mOP) to protect against basal stem rot (BSR) compared to planting conventional oil palm (cOP). CT = current time.

| | | | Year | |
|---|---|---|---|---|
| | | **CT** | **2070** | **2100** |
| | Costs of BSR for cOP | 376 | 789 | 1314 |
| Indonesia | Costs of BSR for mOP | 376 | 676 | 995 |
| | Savings from using mOP | | 113 | 319 |
| | Costs of BSR for cOP | 19 | 115 | 182 |
| Thailand | Costs of BSR for mOP | 19 | 97 | 132 |
| | Savings from using mOP | | 18 | 50 |

## 4. Discussion

According to the data presented herein, the use of mOP would not completely remedy the problem of BSR. The combination of the (a) length of time until the mOP will be available, (b) gradual introduction of mOP to plantations, and (c) likelihood that the mOP will not be 100% effective means that high levels of infection, OP mortality and reduced yields will still occur. Stakeholders may find it informative to consider these possibilities. Nevertheless, research into mOP is ongoing, and major breakthroughs may occur making control very effective.

BSR already has a high incidence at the current time in Indonesia (Figure 1), although it progressed less rapidly than in Thailand under the current scheme. Substantial numbers of healthy OP remained even by 2100 in Indonesia, especially when compared to Thailand. In general, there was a significant reduction in BSR when the planting of mOP was postulated in Indonesia. The number of healthy OP in Indonesia was at least a factor of 10 higher than in Thailand (Figure 2), and there were considerably lower percentages of mortality in Indonesia (Figure 3). The yields were more stable until 2070 in Indonesia but dropped rapidly after 2070 for cOP. mOP allowed the maintenance of high yields for Indonesia (Figure 5), which were more than a factor of 10 higher than for Thailand (Figure 6). The introduction of mOP resulted in low mortalities in Thailand, indicating that the process may be useful for the sustainability of the industry.

The number of OP saved from BSR due to planting mOP in Indonesia by 2070 was ca. 50% of the current total OP in Thailand and the number of OP saved by 2100 was ca. twice the current total of OP in Thailand. The total number of OP saved from death by using mOP in 2070 in Indonesia was ca. 33% of the total current number of OP in Thailand. In addition, the total number in 2100 in Indonesia was approximately equal to the total number of OP currently planted in Thailand (Tables 1 and 3). The yield gain in 2100 from using mOP in Indonesia was ca. three times the total current Thailand yield, and the 2070 figure was ca. 50% of the current Thailand yield (Tables 1 and 3).

**Table 3.** Total area planted with OP (ha $\times 10^6$), number of OP ($\times 10^7$), and yield of palm oil (t yr$^{-1}$ $\times 10^5$) in various continents and countries ($\times 10^6$) [1].

|  | Hectares | Number OP | Yield |
|---|---|---|---|
| Indonesia | 6.50 | 91.0 | 298.3 |
| Malaysia | 4.36 | 61.0 | 209.5 |
| Africa | 1.25 | 17.5 | 24.3 |
| South America | 0.71 | 9.9 | 21.9 |
| Thailand | 0.64 | 9.0 | 18.5 |
| Central America | 0.29 | 4.1 | 12.4 |
| PNG | 0.14 | 2.0 | 5.6 |
| India | 0.08 | 1.1 | 1.0 |

The number of OP without BSR in Indonesia in 2070 by planting mOP was approximately 5% of the total number of OP at the current time, which is not a large saving. However, it is more than the total current number of OP in Central America, PNG, or India (Table 3) and is significant in this respect.

The number in 2100 was approximately 18% of the total initial number of OP in Indonesia, which is a significant number and represents (a) a similar number to the total in Africa, (b) almost twice the number in South America, and (c) much more than in Central America, PNG, or India. For Thailand, the number of OP without BSR by planting mOP in 2070 was approximately 11% of the total initial OP at the current time and represents a significant gain, although careful consideration is required to assess if this would justify the effort of introducing mOP. This value is similar to the total number of OP in India (Table 3) and so is significant in this respect. The number saved from BSR by 2100 was about 33% of the total OP at the current time, which may be worth the effort of developing mOP, although the BSR incidence rate may be too high even when mOP is considered.

The number of OP saved from death in Indonesia in 2070 was ca. 3% of the total number of OP ($910 \times 10^6$), which may appear scant reward for planting mOP. Nevertheless, this number is more than the total number of OP currently in PNG or India and may be worthwhile when considered in this light. Palm oil may become increasingly valuable in the future because of shortages due to climate change, and thus apparently small savings could become worthwhile. The increase in living OP in 2100 using mOP represents 9% of the total Indonesian OP grown currently. This figure is only slightly less than the number of OP in South America or Thailand and is considerably more than that in Central America, PNG, or India (Table 3). The number of OP saved from dying in Thailand in 2070 and 2100 was 11% of the total number of OP in Thailand currently, which is substantial and may indicate that the introduction of mOP would be worthwhile. This value is similar to the number of OP in India currently and ca. 50% of the total current number in PNG.

The savings in yields in 2070 from employing mOP in Indonesia was 3% of the current Indonesian yield and may not appear worthwhile. Nevertheless, it represents more than the current yields of PNG or India (Table 3) and is similar to that for Central America and ca. 50% of the total Thailand yield. The yield saved in 2100 was 20% of the current Indonesian yield and appears a substantial quantity and worth the investment in mOP technology. This saving is much higher than the current yields of Africa, South America, Thailand, Central America, PNG, or India and is ca. 30% of the total current yield of Malaysia. The

saving in yield from using mOP for 2070 and 2100 for Thailand was 5% of the total current yield of Thailand. This is a small gain from planting mOP, although this amount is the same as the current yield of India which may be considered worthwhile.

A companion study is available concerning the effect of climate change on OP in Indonesia, Malaysia, Thailand, and PNG [42]. Only two parameters of OP mortality and yields were assessed, and more countries could be studied compared to the present investigation, which had three parameters, i.e., BSR, mortality from BSR infection, and yields after BSR infection. It was apparent that Malaysia was similar to Indonesia for the parameters considered in reference [42] and BSR may also have similar effects in the two countries. Thailand was again affected very seriously and PNG was intermediate between (a) Malaysia and Indonesia and (b) Thailand. Hence, the combined effects of BSR and future climate will have a detrimental effect on the palm oil industry, and mOP will have a moderately positive effect. However, the improvements may be highly significant given the possibility that the commodity will become scarce in the future because of climate change.

## 5. Conclusions

The costs of BSR in the future to the industry under current economic circumstances are very high for Indonesia and to a lesser extent for Thailand, which has much lower production. The value of using the computer and narrative model described herein have been demonstrated. The economic situation in the future is uncertain, and there may be much less palm oil available due to the effects of climate change. This would tend to make the market cost of palm oil high, allowing for the profitable production of small volumes. The savings from employing mOP would be low for Thailand, making the introduction questionable. World economies will be negatively affected by climate change and it is unclear if the market for palm oil will remain buoyant and stakeholders need to consider carefully the benefits of developing mOP.

**Funding:** This research received no external funding.

**Data Availability Statement:** Data are contained within the article.

**Conflicts of Interest:** The author declares no conflict of interest.

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
