# Peer review of "Future Climate Effects on Basal Stem Rot of Conventional and Modified Oil Palm in Indonesia and Thailand"

_forests, doi:10.3390/f14071347_

Round 1

Reviewer 1 Report

See the comments in the manuscript attached.

Reviewer 2 Report

The Abstract and the Introduction of the paper start with one and the same sentence. One of them should be rewritten.

Please, cite correctly the reference [18] in the paper, or change the numbering.

Line 171, "D was incidence in Indonesia or Thailand." shrink the font.

The Figures' explanations should be more clear and without tautology. Avoid "(a)" and "(b)" from the sentences, or use them by different manner.

Line 191, "CT" is not explaine. 

Unify the label of Figure 2.

I recommend useage of 0.9x107,  instead of 9.0x106, (line 196 and in the rest of the Results, including explanations for Tables).

Line 213, "In Figure 4 the number ..." use a new line.

Line 229, avoid the tautology "more rapidly".

The Tables should be renumbered. Cite and explaine them correctly.

Use "Thailand", instead of "Thai".

Delete blank lines in REFERENCES.

I recommend parts 3.Results and 4.Discussion to be clearly rewritten and the quality of the English language to be checked more precisely. 

Round 2

Reviewer 1 Report

The author has made substantial improvements to the manuscript. 

Reviewer 2 Report

I have no more comments and suggestions for authors.